# The Release of Antimony from Mine Dump Soils in the Presence and Absence of Forest Litter

**DOI:** 10.3390/ijerph15122631

**Published:** 2018-11-24

**Authors:** Karolina Lewińska, Anna Karczewska, Marcin Siepak, Bernard Gałka

**Affiliations:** 1Department of Soil Science and Remote Sensing of Soils, Adam Mickiewicz University in Poznań, ul. Krygowskiego 10, 61-680 Poznań, Poland; 2Institute of Soil Science and Environmental Protection, Wrocław University of Environmental and Life Sciences, ul. Grunwaldzka 53, 50-357 Wrocław, Poland; anna.karczewska@upwr.edu.pl (A.K.); bernard.galka@upwr.edu.pl (B.G.); 3Institute of Geology, Adam Mickiewicz University in Poznań, ul. Krygowskiego 12, 61-680 Poznań, Poland; marcin.siepak@amu.edu.pl

**Keywords:** Sb, solubility, organic matter, waterlogging, MacroRhizon

## Abstract

This study examined the changes in antimony (Sb) solubility in soils, using organic matter introduced with forest litter, in various moisture conditions. Soils containing 12.8–163 mg/kg Sb were taken from the top layers of dumps in former mining sites in the Sudetes, South-West Poland. Soils were incubated for 90 days either in oxic or waterlogged conditions, with and without the addition of 50 g/kg of beech forest litter (FL). Water concentrations of Sb in some experimental treatments greatly exceeded the threshold values for good quality underground water and drinking water, and reached a maximum of 2.8 mg/L. The changes of Sb solubility caused by application of FL and prolonged waterlogging were, in various soils, highly divergent and in fact unpredictable based on the main soil properties. In some soils, the application of forest litter prompted the release of Sb from soil solid phase, while in the others it acted contradictorily. Soil waterlogging resulted, in most cases, in the increased release of Sb compared to oxic conditions, and this effect was enhanced by the addition of forest litter. However, in two soils the presence of forest litter counteracted the effects of waterlogging and diminished the quantities of released Sb.

## 1. Introduction

Antimony (Sb), a potentially toxic metalloid, has recently attracted increasing attention in environmental sciences [1,2,3,4,5]. Particularly strong soil enrichment with antimony was recorded from the sites of contemporary or historical mining and processing of antimony or arsenic ores [3,6,7], with the highest concentrations of Sb occurring in mine dumps. In the area surrounding Xikuangshan, the world’s largest Sb metallurgical complex in China, soil concentrations of Sb reached 7620 mg/kg Sb [8], and in a large area exceeded 5000 mg/kg [9]. Similar or even higher levels were reported from various European antimony mining sites, for instance from Slovakia, where mine wastes (in Čučma) and soils (in Dúbrava) contained over 9000 mg/kg Sb [10]. Álvarez-Ayuso et al. [11] found Sb concentrations in the range of 585–3184 mg/kg in an area impacted by a former stibnite mining in San Antonio, Spain, and Courtin-Nomade et al. [12], reported the mean concentrations of Sb in slags and tailings in the Ouche site in Brioude-Massiac district (France), at levels of 1700 and 5000 mg/kg, respectively. The data from Polish historical Sb mining sites in the Sudetes, published previously, indicated Sb concentrations up to 427 mg/kg [13]. The latter value is much lower than values recorded from the largest antimony mines in the world; however, it is more than three orders of magnitude higher when compared to the value of 0.2–0.3 mg/kg, a global geochemical background value [3]. It is also the highest value ever reported in Polish soils. It should be added that soil concentrations of Sb in Poland to date have been poorly recognized, and the data from national monitoring of farmlands are in the range of 0.06–1.03 mg/kg [14].

Strong enrichment of soils in Sb can pose a considerable risk to the environment. It should be stressed, however, that real hazards will depend on Sb solubility rather than on its total concentrations in soils. Antimony can exist in the environment in four oxidation states (−III, 0, III, and V) of which Sb(III) and Sb(V) are the most frequently occurring species. The general order of their toxicity is: Sb(III) > Sb(V) > organoantimonials [10,15]. Conversion of one Sb species into another (Figure 1) can be mediated by microbial transformations [16,17,18,19].

Although antimony is usually considered poorly soluble in the environment, it can be released from enriched rock and soils, thus leading to pollution of natural water and enhanced uptake by aqueous and terrestrial plants. Several authors reported highly elevated concentrations of antimony in water draining the dumps in abandoned mine sites [20,21,22], reaching up to 0.75 mg/L in water of Prestea, a gold mining town in Ghana [23], and 11.4 mg/L in stream water receiving seepage from ore residues in Chinese Xikuangshan [24], while 20 μg/L has been set by the WHO as a recommended value.

Once released into natural water or soil pore water, antimony can be taken up by aquatic or terrestrial plants, thus entering the food chain. Algae can take up more than 10 mg/kg of Sb from polluted natural water [25,26]. Telford et al. [21] reported concentrations 96–212 mg/kg Sb in aquatic autotrophs present in the stream adjacent to the Hillgrove Mine in Australia.

Considerable uptake of Sb by terrestrial plants was reported from mine sites by various authors [27,28,29,30,31]. Particularly high Sb concentrations, above 200 mg/kg or even 1000 mg/kg, were recorded, for instance, in the aboveground parts of sweet yarr4ow, maidenstears [27], wild strawberry, and dandelion [31].

Antimony is usually considered relatively immobile in soils as it is bound strongly to iron hydroxides, similarly to arsenic (As). Because of their identical outer orbital electron configuration, Sb and As display a lot of similarities in their environmental behaviour. The conditions when As can be released from soil solid phase into solution have already been widely examined. It has been proved that its release into soil pore water may be caused by the presence of anions, such as phosphates, or by organic compounds that compete with arsenates for oxide sorption sites. The changes in soil pH or redox potential will also affect the solubility of As in soils. Similar relationships have not been confirmed in the case of Sb. Only recently, researchers have studied the factors that determine Sb release into pore water, and its biochemistry has been revealed to be more complex compared to As [1,2,32,33]. Sb can be released from soil solid phase via weathering and leaching. Mobilization of Sb from the rocks or soils in mineralised areas occurs usually as an effect of oxidation and dissolution of Sb bearing primary minerals. For instance, stibnite (Sb_2_S_3_) dissolution at active mining sites was reported to produce Sb concentrations up to 55 mg/L of Sb in solution [34]. In soil pore water, under strongly acidic conditions, Sb exists in cationic forms, as Sb(OH)^+2^ or SbO^2+^, but in neutral or basic pH it forms anions [Sb(OH)_6_]^−^ or SbO_3_^−^ [33]. The process of Sb release into pore water is usually followed by its relatively rapid removal from solution due to adsorption of anionic forms and their co-precipitation with amorphous iron or calcium hydroxides [35]. Adsorbed Sb may be again released from soil, and the factors and conditions that govern this process are only partly similar to those recognized for As. The oxidation state, pH, and redox potential are of crucial importance. Antimony is mainly present in soils as Sb(V), and usually remains associated with Fe hydroxides in the form of Sb(V) even under reducing conditions [1,2]. Sb(III) may be rapidly oxidized to Sb(V) by amorphous forms of Fe and Mn oxyhydroxides in the pH range 5–10 [36] and this process can, in fact, contribute to mobilisation of Sb species in alkaline soils. Several studies confirmed that Sb may also be released from soil through reductive dissolution of Mn and Fe (hydro)oxides under waterlogged conditions [2,37]. The transformation of Sb(V) to Sb(III), however, usually counteracts this effect, because Sb(III) sorbs more strongly and extensively to hydroxides than Sb(V) [1,2,15,36].

The effects of organic matter on Sb sorption or release from soils are ambiguous [38]. Contrary to the observations reported for As, competition with soluble organic acids or particulate organic matter to sorption on hydroxides has not been proved for Sb [2]. However, considerably high solubility of Sb was reported by Gerritse et al. [39] after soil amendment with sewage sludge, and Clemente et al. [40] stated that application of green waste compost mulch to soil increased organic carbon and Fe in soil pore water, which in turn increased As and Sb mobilization. Hockmann et al. [5] explained the seasonal fluctuations in Sb leaching from non-waterlogged calcareous soil by fluctuations in dissolved organic carbon (DOC) and bicarbonate concentrations, and, based on a microcosm experiment, they proved that the addition of lactate in reducing conditions can lead to a slow increase in Sb solubility. The same authors stressed therefore that application of organic fertilizers, such as manure, to soils prone to waterlogging is not recommended as it may lead to the mobilization of hazardous Sb(III) upon reductive dissolution of their host phases. Similarly, biochar used as a soil amendment caused reduction of lead mobility and bioavailability, while antimony got mobilized [41]. On the other hand, however, various authors reported strong binding of neutral Sb(III) species such as Sb(OH)_3_ to humic acids, particularly in highly organic acid soils [2,33,42,43,44,45] and concluded that interactions with humic compounds can significantly reduce Sb mobility. Several mechanisms for Sb(III) binding with functional groups of humic matter, mainly phenolic, carboxylic, and hydroxyl-carboxylic groups, were proposed, including ligand exchange with the Sb centre, and formation of negatively charged Sb complexes with carboxylic groups. Moreover, chelation, H-bridges and the presence of cationic metals may contribute to the stabilization of the Sb(III) binding.

All the contradictory statements concerning the effects of organic matter on the release of Sb from soils prompted us to examine the solubility of Sb in dump soils as affected by the presence of organic matter derived from forest litter. This knowledge will be of particular importance from the standpoint of environmental risk assessment which is an approach required by law in various countries, including Poland [46,47]. Organic compounds with potential chelating properties produced during forest litter transformation can considerably affect the solubility of Sb. The release of Sb from the solid phase can also be intensified by locally occurring or reducing conditions. Such effects have been proved in the case of As [48,49]. The present study was aimed to determine the changes in Sb solubility in various mine dump soils treated with forest beech litter, and incubated in various moisture conditions, including waterlogging.

## 2. Materials and Methods

### 2.1. Sampling Sites and Soils

Soil material (six large samples, each of ca. 100 kg) was collected from the top layers of mine dumps in five areas of arsenic and antimony ore mining (Złoty Stok (ZS), Dębowina (DB), Dziećmorowice (DM), Radzimowice (R), and Srebrna Góra (SG)), situated in different parts of the Sudetes (Figure 2).

Geological settings and mining history of the five sampling sites were described by Karczewska et al. [50]. Generally, mineralization occurs either in serpentinised dolomitic marbles and calc-silicate skarn-type rocks (Złoty Stok) or as vein-type sulphide hydrothermal intrusions (other sites) formed in dislocation zones of various geological structures. The ZS1, ZS2, DB, R, and SG dumps are surrounded by forests, while DM, sparsely covered by trees, is adjacent to fields and meadows. They are all situated in hilly or mountainous landscapes, typically in the valleys drained by streams that continue through the villages and in some cases supply water to homesteads. Four of the soils, relatively rich in arsenic, were previously examined on release of As [49] and described in more detail. Two samples (ZS1 and ZS2) represented the largest mine dump (Orchid Dump, 2.3 ha) in Złoty Stok. The coarse fractions of soils, i.e., the fractions larger than fine gravel (>5 mm), were discarded prior to the experiment, and the texture of material prepared in that way was determined by a combined sieve and hydrometer method. Basic chemical properties were examined in aliquots of fine soil using standard methods [51].

The concentrations of Sb extractable with aqua regia, close to real total concentrations [13] were determined after microwave assisted digestion. Additionally, the pools of Sb, extractable with 1 M NH_4_NO_3_ and 0.01 M CaCl_2_, were determined, and five operationally defined fractions of Sb were separated in soils using a sequential extraction method dedicated particularly for As [52]. The fractions are believed to represent non-specifically-bound (I), specifically-bound (II), amorphous hydroxide-bound (III), crystalline hydroxide-bound (IV), and the residual forms (V) of Sb. This procedure was found to be a suitable scheme for evaluating the possible mobilization processes from the samples contaminated by ore processing waste [53]. The fractions I–III, presented in Table 1, are of particular importance from the standpoint of Sb solubility. All the measurements of Sb concentrations in digests and extracts were determined by ICP-MS (8800 QQQ, Agilent Technologies, Santa Clara, CA, United States). All reagents were of analytical grade (Merck, Darmstadt, Germany). Validation of this method involved the analysis of the certified reference material CNS 392 (Sigma Aldrich, Darmstadt, Germany) as well as the procedure of standard addition applied to selected samples. All extractions were carried out in triplicate unless stated otherwise.

### 2.2. Incubation Experiment

After discarding the coarse fractions larger than fine gravel (>5 mm), soils were incubated for 90 days in 1-kg pots at two levels of soil moisture: 80% of water holding capacity and 100% of maximum water capacity, with and without addition of beech forest litter (FL) (50 g/kg). Soil pore water was collected after 2, 7, 14, 28, and 90 days with MacroRhizon suction samplers that proved to be suitable for research focusing on trace elements [47,54]. Pore water was examined for Sb concentrations, pH and Eh. Additionally, dissolved organic carbon (DOC), as well as As, Mn, and Fe were determined in pore water after a prolonged time of incubation [49]. The experiment was carried out in three replicates, and the data presented in this paper illustrate the mean values and confidence intervals at *p* = 0.95.

### 2.3. Forest Litter

Forest litter (FL) collected from a 60-year-old beech stand was air-dried, crumbled, sieved to 1 cm, and analysed prior to being used in the experiment. Analyses involved determination of total organic carbon, dissolved organic carbon (DOC) extracted in hot and cold water (according to the method by Gregorich et al. [55]), pH in water suspension, and fractional composition of humic substances (as described by Karczewska et al. [49]). Additionally, humic acid fraction was extracted from FL using a method recommended by International Humic Substances Society [56] and examined by Fourier transform infrared spectroscopy (FTIR, Bruker Vertex 8, Billerica, MA, United States) in order to illustrate the abundance of various functional groups.

### 2.4. Pore Water Analysis

Pore water samples were filtered through a 0.45-μm membrane and analysed immediately after collection. The concentrations of Sb were determined by ICP-MS. Validation of analysis was checked for selected samples by standard addition. Additionally, in pore water collected from waterlogged samples, the concentrations of Mn and Fe were measured (by ICP-MS) and redox potential Eh was determined potentiometrically. The pH values of pore water were measured potentiometrically, and the concentrations of dissolved organic carbon (DOC) were determined spectrophotometrically by a standard method with the Merck 1.14878.0001 test [49].

### 2.5. Statistics

Significance of differences among the treatments and various periods of incubation time were analysed by a two-way ANOVA, followed by post hoc multiple comparison of means carried out by a Tukey’s test (*p* <0.05). Statistical analysis was performed using Statistica software, version 12.0 (Dell Inc., Round Rock, TX, United States).

## 3. Results

### 3.1. Soils and Forest Litter

Dump soils contained a relatively high contribution (45–80%) of skeletal fractions (>2 mm), as roughly determined in the field. The material used further in the incubation experiment, after discarding grains which were >5 mm, contained 25–48% fine gravel, and its fine fractions represented loams, sandy loams, and loamy sands (Table 1).

All soils, despite their initial stage of development, contained relatively high amounts of organic carbon (Corg.), in the range 13.2–51.4 g/kg. The differences of pH values and chemical composition reflected the varied mineralogy of metal-bearing rocks and the zones that host mineralization. Total concentrations of Sb ranged from 12.8 to 195 mg/kg, and were much lower than those reported from various Sb mine sites in the world. Unexpectedly low (12.8 mg/kg) was the concentration of Sb in DB soil collected from the stibnite mine dump. Sb extractability, determined with 1 M NH_4_NO_3_ and 0.01 M CaCl_2_, was, in most samples, below 0.2 mg/kg, except for DM soil where it was higher, but did not exceed 0.8 mg/kg. Sequential extraction showed an extremely low contribution of fraction I (0.2–1.8% of total Sb), slightly larger amounts of Sb in fraction II (1.2–6.7%), and a high percentage of Sb in fraction III (25.0–78.3%), particularly in the DM and SG soils (Figure 3).

Forest litter contained 418 g/kg of Corg. Cold and hot water-soluble organic compounds represented 0.94% and 3.76%, respectively, of total carbon (Table 2). The data obtained from the fractionation of organic matter revealed a fairly large share of fulvic acids (45.7% of total Corg.), and a lower share of humic acids (32.7%). FTIR spectrum of separated humic acids indicated their considerable reactivity, related to various kinds of functional groups (Figure 4).

### 3.2. Sb Release into Pore Water

The concentrations of Sb in soil pore water collected from the experiment were, depending on soil, conditions and time of incubation, in a very broad range: from ~1 μg/L to over 2.5 mg/L. The latter maximum value is much lower than that described by Ashley et al. [34], who reported dissolution of stibnite producing up to 55 mg/L of Sb in solution in mining sites. On the other hand, however, it is a value that is two orders of magnitude higher than the that of the Polish standard for satisfactory-quality underground water (25 μg/L).

#### 3.2.1. Sb Release in Oxic Conditions

A release of Sb from soils incubated in oxic conditions without forest litter (Figure 5: 0/80%) differed strongly among the soils. The lowest Sb concentrations, ~1 μg/L, were present in the pore water of acidic DB soil, and the highest, over 200 μg/L—in the soil DM, at pH 6.5. This relationship well reflects the dependence of Sb solubility on pH.

Similar to As, Sb remains usually poorly soluble in the pH range 4–6, and its solubility increases along with increasing pH in neutral and alkaline conditions. During incubation, there is no clear trend of changing Sb concentrations in pore water of untreated soils (Figure 5). In some soils and time periods the changes were statistically insignificant, but in some cases a temporary, significant increase (DB, R) or decrease (ZS2, DM) of Sb solubility was reported. Comparison of the beginning and end of incubation (2 days vs. 90 days) indicates either no differences or contradictory changes in Sb concentrations in pore water, i.e., a small increase in ZS1, and a decrease in R and SG. Those changes seem to be, however, of negligible importance, and probably reflect a natural variability of soil properties.

The results indicate further that soil treatment with FL in oxic conditions (Figure 5: FL/80%) did not result in any convergent effects on Sb release either. Sb mobilization was reported at the beginning of incubation in the soils R and SG, and at the same time, the application of FL caused immobilization of Sb in the soil DM. In the long time period (90 days), the treatment with FL resulted in reduction of Sb solubility in the soils ZS1, ZS2, and DM, but an opposite effect was still observed in the soil R.

#### 3.2.2. Sb Release in Flooded Soils

Soil waterlogging during the incubation resulted also in divergent, though in most cases statistically confirmed, changes of Sb release compared to oxic conditions (Figure 6: 0/100% vs. 0/80%). In short time periods, the concentrations of Sb in soil pore water were considerably, up to over three-fold higher in waterlogged treatments than in oxic ones, with the only exception being the SG soil which is very rich in carbonates.

Application of forest litter followed by incubation in waterlogged conditions (Figure 6: FL/100%) resulted in some soils (R, DM, SG) in a significant, strong increase in Sb release, particularly at the beginning of incubation. This effect turned out to be only temporary, with the maximum after 2 days or 7 days, and then the Sb concentrations in pore water of FL-treated, waterlogged soils started to decrease, so that after 90 days they were comparable to those in untreated soils. Unlike that effect, the reduction of Sb release was reported under waterlogging in the ZS1 and ZS2 soils treated with FL.

## 4. Discussion

The experiment revealed that the effects posed by organic matter and various redox conditions on the solubility of Sb differ radically depending on soil origin and properties. This observation is coherent with the statement provided by various authors that Sb behaves differently from As. The latter was mobilized from soils, independent of the soils’ properties, after the addition of external organic matter in the form of beech forest litter [49]. Several other authors also wrote that the tendency of Sb to be leached into soil solution or remain sequestered in solid phases, particularly in partly weathered dump material, depends on various factors, including the primary host minerals and the conditions governing precipitation of secondary Sb-hosting phases [57,58,59]. Furthermore, particularly important are the effects of desorption vs. resorption on the components of the soil solid phase, which are strongly related both to pH and redox conditions [1,3]. The pH-dependence of Sb solubility in soils, at least in the range of acidic pH, showed up well in our experiment. The lowest concentrations of Sb in pore water of untreated soils incubated in oxic conditions were found in the most acidic soil, DB, with an initial of pH 3.7 (with the lowest total Sb), as well as in the soil R (at pH 4.8) (Figure 5). These two soils also contained the lowest amounts of NH_4_NO_3_ and CaCl_2_-extractable Sb (Table 1, Figure 3).

Much higher was, in turn, the solubility and extractability of Sb in neutral and alkaline soils. The differences between the soils, and a particularly high release of Sb from the soil DM indicate however that, apart from pH, there were some other essential factors that governed the rate of Sb mobilization. Moreover, the time-dependence of Sb release during incubation in oxic conditions (Figure 5) varied among the soils and did not follow any common pattern. A significant decrease in pore water Sb, observed in the soils R and SG after the longer time of incubation, can possibly be attributed to re-precipitation processes and the formation of secondary insoluble minerals. Those two soils were relatively rich in Sb, and their Sb concentrations in pore water were sufficiently high to take such processes into consideration. Wilson et al. [2] stressed that precipitation and co-precipitation processes are only important when Sb concentrations are high enough to initiate precipitation. In such circumstances, secondary Sb minerals can be formed and control dissolved concentrations, whereas in soils with lower total Sb concentrations, this is an adsorption mechanism that governs Sb mobility. In the case of SG soil, which is very rich in calcium carbonate, an initial release of Sb into soil solution could likely be followed by re-precipitation of poorly soluble Ca-based minerals. The anions [Sb(OH)_6_]^−^ and SbO_3_^−^, i.e., the predominant water soluble Sb species at neutral pH, can be co-precipitated with Ca^2+^ ions, forming insoluble and stable calcium antimonate Ca[Sb(OH)_6_]_2_ or romeite Ca_2_Sb_2_O_7_ [7,60]. Such a hypothesis should be checked in a more advanced study performed on soil solid phase after incubation. The concept of re-precipitation with Ca^+2^ ions does not explain, however, the removal of Sb from pore water during incubation of the soil R which was more acidic. It does not provide an explanation, either, of the opposite behaviour of Sb in the soils DM and ZS2, in which the initial decrease of Sb concentrations in pore water of untreated soils was followed by a further increase so that the final Sb concentrations in water did not differ from the initial values (Figure 5).

Particularly high solubility of Sb in the untreated soil DM distinguishes this soil among the others. The total amounts of Sb released from this soil during incubation, calculated based on Sb concentrations and the volumes of removed pore water, turned out to be in all the experimental treatments a great deal higher than those released from the other soils (Table 3). It is worth mentioning that As solubility in this particular soil was also extraordinarily high [49]. A specificity of that soil cannot be explained based on its chemical properties alone and, in further study, a mineralogical approach will be necessary, focussing on the potential presence of such minerals as mopungite, brandholzite, or bottinoite that have appreciable solubility [2,34].

As reported above, the introduction of FL into soils did not impose any consistent effect on the solubility of Sb. In the oxic conditions (Figure 5: FL/80%) that effect was either negligible or, in more cases, the solubility of Sb in soils decreased compared to untreated soils. Reduction of Sb solubility can probably be attributed to the formation of binds with functional groups of humic acids that were present in relatively well humified organic matter. The lack of any effects could, in fact, be expected, as for a long time Sb was considered indifferent to organic matter. Though, the latest sources confirmed that Sb may form an association with humic substances, and neutral Sb(III) species can readily bind to humic acids [2,33,42]. Moreover, Sb(III) binds comparatively more strongly than As (III) [33].

It should be stressed, however, that an opposite effect was observed in the soils R and SG, which were both relatively rich in Sb. Application of forest litter to those two soils induced an intensive Sb mobilization (Figure 5). The concentrations of Sb in FL-treated R and SG soils were higher than those in untreated soils, and significant differences persisted until the end of incubation (90 days). These two soils differed considerably from one another in terms of their basic properties, including pH (4.8 and 7.8), with the only common feature being the small content of native organic matter (Corg.: 18.0 and 13.2 g/kg). It seems impossible to identify a common factor that could have been responsible for Sb mobilization by forest litter in these two soils. It could be hypothesized that mechanisms were involved such as the competition between Sb-bearing ions and dissolved low molecular organic compounds present in FL for the sorption sites on iron (hydro)oxides, or the possible chelation of Fe that might have triggered Sb mobilization and prevented it from re-sorption. A small increase of pH caused by FL application to soils (Appendix A) could not be responsible for the significant changes in Sb solubility.

The matter was even more complex when considering the results obtained in the conditions of waterlogging. Generally, a release of Sb from flooded soils (Figure 6, Table 3: 0/100%) that was, except for the soil SG, significantly higher than in oxic conditions (0/80%), was rather not expected based on the literature. However, Frohne et al. [61] reported the higher concentrations of Sb in pore water of waterlogged soils compared to oxic conditions. Reduction of Sb(V) to Sb(III), likely to take place in reducing conditions, would in fact result in removal of Sb from soil pore water, as Sb(III) is much more strongly sorbed by iron oxides compared to Sb(V) [33]. Hockman et al. [5] reported, therefore, decreased leaching of Sb from waterlogged soils. The effect of increased Sb solubility in our experiment cannot be attributed to reductive dissolution of iron (hydro)oxides either, as it was observed in a relatively short time (2 days) whereas Fe-reducing conditions develop slowly, as in stagnic, gleyic soils subjected to waterlogging [37]. A slowly progressing decrease in redox potential was confirmed in our study by the measurements of Eh (Appendix A). More efficient mobilization of Sb from soil solid phase in flooded conditions can possibly be explained by a simple shift of equilibrium in dissolution processes of moderately soluble Sb-hosting minerals [2].

The release of Sb from FL-treated soils incubated in flooded conditions (Figure 6: FL/100%) was in ZS1 and ZS2 reduced compared to untreated soils, but Sb solubility increased strongly in all the other soils. That effect was only temporal, and did not last beyond day 14. The concentrations of Sb in pore water of the soil DM rose to the extremely high values above 1 mg/L. This effect, upon soil flooding, should not be attributed to a rapid oxidation of Sb (III) to Sb (V) by amorphous Fe and Mn (hydro)oxides, which is theoretically possible in the pH range 5–10 [1]. We would rather explain it by microbiologically mediated processes of reductive iron (hydro)oxides dissolution, occurring locally, where the easily available fraction of organic matter acts as an electron donor, stimulating the rate of Fe(III) reduction or intensifying a reduction of Sb, as confirmed recently by various authors [17,18]. Similarly, Rowland et al. [62], who studied mobilization of As from soils, suggested that certain types of organic matter may accelerate its release by acting as an electron shuttle. Biomethylation processes should probably also be considered [2,17,61].

To sum up, it should be stressed that the release of Sb into pore water from mine dump soils does not follow any common pattern and, unlike the release of As, cannot be explained nor predicted without undertaking comprehensive mineralogical studies, and possibly also biochemical studies, of each individual case. The results of soil extraction with 1 M NH_4_NO_3_ provide, however, useful information on the likelihood of possible Sb mobilization. In certain conditions, in particular in the presence of biodegradable organic matter, antimony can be released into water in the amounts that might present a threat to the environment and to the health of local inhabitants.

## 5. Conclusions

This study confirmed the relatively poor solubility of antimony in soils, with as little as 0.22% of total Sb being the maximum amount released into pore water during the 90-day incubation. However, the concentrations of Sb in pore water in some of experimental treatments greatly exceeded the threshold values for a satisfying quality of underground water (25 μg/L) and the WHO standard for drinking water (20 μg/L), thus posing the considerable risk to the environment and to humans.

The effects of various processes involved in Sb release from mine dump soils and immobilization are often contradictory and result in opposing changes in Sb solubility. The presence of forest litter, as well as waterlogged conditions, can significantly affect the solubility of Sb in soils; however, these effects differ strongly among the soils. In some soils, the application of forest litter prompted the release of Sb from soil solid phase, while in the others it acted contradictorily. Soil waterlogging generally resulted in increased release of Sb compared to oxic conditions, and this effect was enhanced by addition of forest litter. Though, in some soils, represented in this study by ZS1 and ZS2, the presence of forest litter can counteract the effects of waterlogging and diminish the amounts of released Sb.

The release of Sb from soils into pore water under oxic conditions apparently corresponds with the results of extraction with 1 M NH_4_NO_3_. Neither those results, however, nor Sb speciation revealed by the sequential extraction, reflect the extent of Sb release under waterlogged conditions or as influenced by the addition of forest litter. High extractability of Sb with 1 M NH_4_NO_3_ should be, however, considered as an important factor that indicates the likelihood of particularly intensive leaching of Sb from soil under flooding or/and in the presence of biodegradable organic matter. Such a case was exemplified in our study by the soil collected from the DM dump. An explanation of the mechanisms involved in Sb mobilization from that particular soil requires a more advanced examination.

## Figures and Tables

**Figure 1 ijerph-15-02631-f001:**
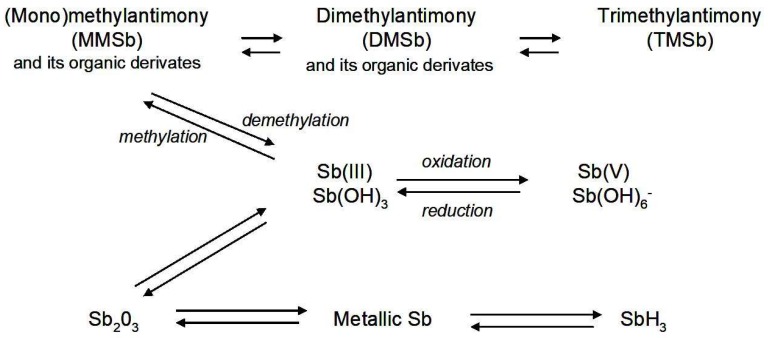
Microbial transformations of antimony (Sb)—a simplified scheme, from [17], which has been modified.

**Figure 2 ijerph-15-02631-f002:**
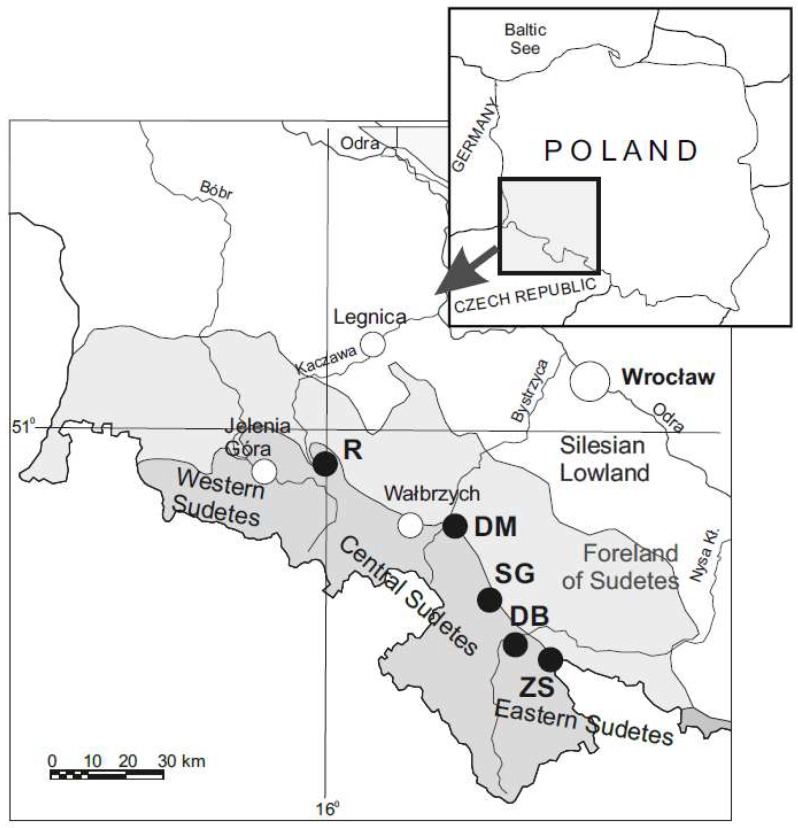
Location of sampling sites. For a closer description of geological settings and mining history see Karczewska et al. [50]. ZS: Złoty Stok; DB: Dębowina; DM: Dziećmorowice; R: Radzimowice; SG: Srebrna Góra.

**Figure 3 ijerph-15-02631-f003:**
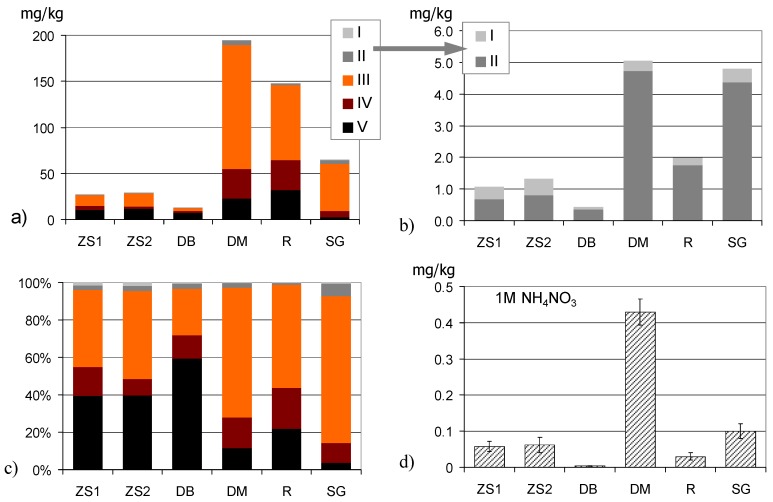
The amounts of Sb extracted from soils in the sequential extraction according to Wenzel et al. [52] (diagrams **a**–**c**), as well as in the single extraction with 1 M NH_4_NO_3_ (diagram **d**). Error bars presented for extraction with 1 M NH_4_NO_3_ stand for standard deviation of three replicates.

**Figure 4 ijerph-15-02631-f004:**
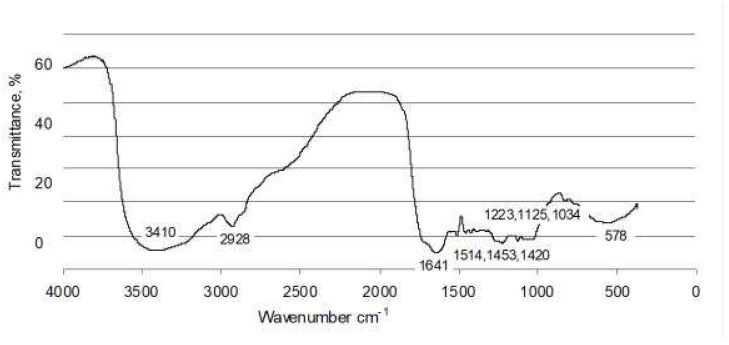
DRIFT (Diffuse Reflectance Infrared Fourier Transform) spectrum of humic acids extracted from forest litter (FL).

**Figure 5 ijerph-15-02631-f005:**
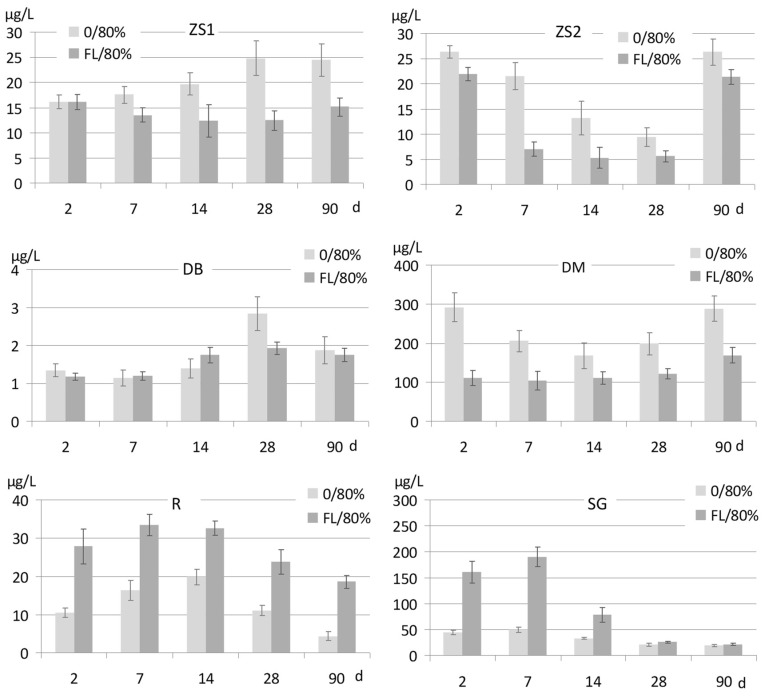
Sb concentrations in pore water of soils incubated at the moisture level of 80% of water holding capacity without addition of forest litter (0/80%) and with beech forest litter (FL/80%).

**Figure 6 ijerph-15-02631-f006:**
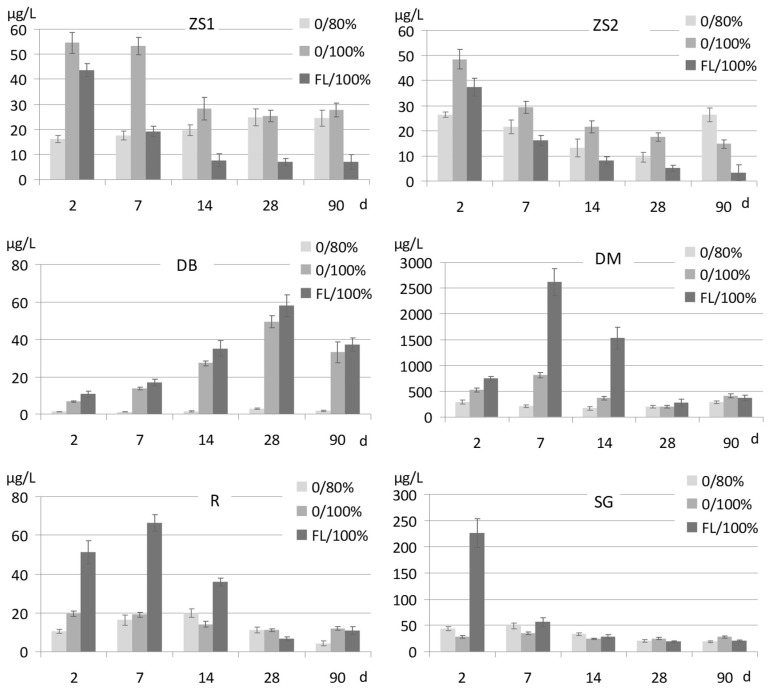
Sb concentrations in pore water of soils incubated at the moisture 100% of maximum water capacity without addition of forest litter (0/100%) and with beech forest litter (FL/100%), compared with the treatment 0/80%.

**Table 1 ijerph-15-02631-t001:** Soil origin and basic properties used in the experiment (the mean values of three replicates). Chemical properties determined in fine soil (<2 mm).

Parameter		Soil					
Symbol		ZS1	ZS2	DB	DM	R	SG
Origin	Locality	Złoty Stok		Dębowina	Dziećmorowice	Radzimowice	Srebrna Góra
	Mined ores	Au and As		Sb	polymetallic		
Skeleton (>2 mm)—on site, %		80	50	45	60	60	55
Fine gravel (2–5 mm)		45	25	36	48	32	27
Sand (0.05–2 mm)	in fine soil, %	68	76	41	75	73	70
Silt (0.002–0.05 mm)		29	22	52	23	22	22
Clay (<0.002 mm)		3	2	7	2	5	8
Textural group of fine soil (USDA)		SL	LS	L	LS	SL	SL
Corg., g/kg		14.1	16.3	51.4	29.5	18.0	13.2
N total, g/kg		1.05	1.23	3.28	1.81	1.22	1.12
pH (H_2_O)		7.1	7.6	3.7	6.5	4.8	7.8
CaCO_3_, %		0.1	1.3	-	-	-	7.6
Fe_ox_, g/kg		28.5	23.9	7.0	3.5	23.7	5.5
Total As (aqua regia)		45,500	50,000	68	196	12,150	56
Total Sb (aqua regia)		27.3	29.5	12.8	195	148	65
Sb extracted with 1 M NH_4_NO_3_	mg/kg	0.06	0.06	<0.01	0.43	0.03	0.10
	% of total	0.22	0.20	<0.08	0.66	0.02	0.06
Sb extracted with 0.01 M CaCl_2_	mg/kg	0.16	0.15	<0.02	0.73	0.16	0.08
	% of total	0.59	0.51	<0.16	1.12	0.11	0.05

ZS: Złoty Stok; DB: Dębowina; DM: Dziećmorowice; R: Radzimowice; SG: Srebrna Góra. USDA (United States Department of Agriculture). SL: sandy loam, LS: loamy sand, L: loam. Corg. – organic carbon. n.d.—not determined.

**Table 2 ijerph-15-02631-t002:** Basic properties and fractional composition of beech forest litter ^1^.

Parameter	Unit	Mean Value
Total organic carbon	g/kg	418
pH (H_2_O)	-	5.95
N total	g/kg	17.4
DOC (cold water)	g/kg	3.95
DOC (hot water)	g/kg	15.7
Fulvic fraction	% of total C	45.7
Humic fraction		32.7
Non-extractable C		21.6

^1^ For details concerning fractionation of humic substances—see: Karczewska et al. [49]. DOC: dissolved organic carbon.

**Table 3 ijerph-15-02631-t003:** The amounts of Sb released from soils into pore water.

Soil	Sb Released, μg/kg				Sb Released, % of Total Sb			
	0/80%	FL/80%	0/100%	FL/100%	0/80%	FL/80%	0/100%	FL/100%
ZS1	11.3	7.3	20.6	7.0	0.04	0.03	0.08	0.03
ZS2	11.4	8.4	12.4	4.8	0.04	0.03	0.04	0.02
DB	0.9	0.8	19.7	22.7	0.01	0.01	0.16	0.18
DM	130	72.8	278	424	0.07	0.04	0.14	0.22
R	4.2	11.5	8.6	12.9	<0.01	0.01	0.01	0.01
SG	13.2	29.2	18.2	25.6	0.02	0.04	0.03	0.04

FL: forest litter.

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
