# Peer review of "The Release of Antimony from Mine Dump Soils in the Presence and Absence of Forest Litter"

_ijerph, 2018, doi:10.3390/ijerph15122631_

Reviewer 1 Report

This manuscript examined the changes in Sb solubility in 6 soils as induced by organic matter introduced with forest litter, in various moisture conditions. I can understand the usefulness of the research. However, the method and discussions. There are parts where explanation is missing about the method and discussion. There are also some careless mistakes. Details are described below.

p.3  l.125 "hydroxide-bound (IV)" => hydroxide-bound (III)?

Table 2 Can not show the ratio of sand or silt? It is general information on the characteristic values of soil and it is very important to compare elution behavior.

p.3  l.175 Corg <= Organic carbon content(g-C/kg) ?

p.8  Figure 5  the moisture 1000% => 100%

p.8  l.238 "This effect turned out to be only temporary・・・"

p.8  l.252 "The pH-dependence of Sb solubility in soils, at least in the range of acidic pH, showed off well in our experiment."

p.10  l.320 " A slowly progressing decrease in redox potential was confirmed in our study by the measurements of Eh in selected treatments, not reported here."

Why is not the change in pH or redox potential(ORP) indicated? Do these values change with extraction time? As it is important to understand the results, please indicate the measured value of the solution after extraction.

Author Response

Dear Reviewer,

thank you for all your comments. Please find below our answers.

All the missing explanation about the methods have been provided, and discussion has been improved. Indicated typographical errors and other  mistakes have been corrected. The details are described below.

p.3 l.125 "hydroxide-bound (IV)" => hydroxide-bound (III)?

à  It  was, obviously, a typographical  error. It has been corrected  :  

Table 2 Can not show the ratio of sand or silt? It is general information on the characteristic values of soil and it is very important to compare elution behavior.

à The share of clay fraction was originally given as a crucial parameter related to soil sorption properties. Though, we agree with Reviewer 3 that  the shares of sand and silt fractions in earthy soil can be informative in respect of mechanical washing off. Therefore, the required data have been included into Table 2.

p.3 l.175 Corg <= Organic carbon content(g-C/kg) ?

à  An abbreviation Corg (without the space - a typo) has been replaced by “organic carbon”  

p.8 Figure 5 the moisture 1000% => 100%

à  As above.  A typo.  It has been corrected:  100%

p.8 l.238 "This effect turned out to be only temporary・・・"

à The sentence has been checked by proofreading and has not been changed.

p.8 l.252 "The pH-dependence of Sb solubility in soils, at least in the range of acidic pH, showed off well in our experiment."

à The sentence has been  changed by proofreading into:  "The pH-dependence of Sb solubility (..), showed up well in our experiment."  (P.13, L.315)

p.10 l.320 " A slowly progressing decrease in redox potential was confirmed in our study by the measurements of Eh in selected treatments, not reported here. Why is not the change in pH or redox potential (ORP) indicated? Do these values change with extraction time? As it is important to understand the results, please indicate the measured value of the solution after extraction.

à As stated in the text, the measurements of pore water Eh were performed directly after extraction. For technical reasons, however, the measurements could not be carried out in some treatments / cases. Additionally, the final measurement was delayed for 3 days, which could have  - to some extent - affected the results. The data obtained from 3 replicates differed considerably, and related values of standard deviation SD were in the ranges: 9-43 mV for Eh, and 0.02-0.31 for pH. Despite these facts, the general tendencies were in most cases clearly visible. Therefore, we decided to present the detailed data (with related comments) in Supplementary materials (Table S.1.).

Reviewer 2 Report

Dear Authors, 

Please find the specific comments on methodology and results to improve the quality of the presentation. Some minor comments and suggestion added to the attached file.

Line 42: Please rate more accurately that 427mg Sb per kg soil considers as moderate or high or very high concentration?

Line 103-106:  How large? You may indicate the size of the soil sample in volume or weight? What kind of devise or methods you used to take the soil from the field and did you choose the soil sampling spot randomly?

Author Response

Dear Reviewer,

thank you for all your comments. Please find below our answers.

All the editorial changes indicated in the pdf version of the text have been made, and the questions (P. 1, L.42;  P.3, L.103) have been addressed. Proofreading and editing of English language and style has been made by a professional editing service.

L.42.   Please rate more accurately that 427 mg Sb per kg soil considers as moderate or high or very high concentration?

à The following text has been introduced (P.2, L.48-53):

The latter value is much lower than those recorded from the largest antimony mines in the world, however it is more than three orders of magnitude higher when compared to 0.2-0.3 mg/kg, a global geochemical background [3]. It is also the highest value ever reported from Polish soils. It should be added that soil concentrations of Sb in Poland have been up to now poorly recognized, and the data from national monitoring of farmlands are in the range 0.06 -1.03 mg kg-1 [14].

L.103   How large? You may indicate the size of the soil sample in volume or weight

à A required information has been added (P.5, L.137):

“each of ca. 100 kg, “

Figure 2. the 0,1 to the 0.1 and for the rest of highlighted numbers in Y axis

à Done

Figure 3.  Transmittance à  Is it the Y axes caption? if so then move it to the left side not on top left corner

à Done

Reviewer 3 Report

Antimony (Sb) is a chemical element classified has a priority pollutant by the European Union. Sb pollution has being increasing in recent years due to intensive mining and metallurgical activities and industrial emissions. Research on ecological risk levels from Sb and on release of Sb from soils is needed. The manuscript “The Release of Antimony from Mine Dump Soils in the Presence and Absence of Forest Litter” aimed to evaluate the solubility of Sb collected in mine dump soils, in Sudetes (SW Poland), in the presence / absence of forest beech litter. The study is pertinent but the manuscript presents several issues, specifically: 1) Introduction is extensive but does not present properly Sb, the environmental burden associated to Sb, for example the release to water and accumulation in organisms, or the contextualization of the Sb pollution in Poland. The Sb oxidation states need to be better explained (eventually through a scheme) as well as the open questions / hypothesis associated to the effects of the microbe-Sb interactions on Sb sorption or release from soils. 2) The Material and Methods section needs to be improved. 3) The results and discussion need to be explained according more from edaphic and environmental information from collecting sites.

Author Response

Dear Reviewer,

thank you for all your comments. Please find below our answers.

1. Proofreading and editing of English language and style has been made by a professional editing service

2. Additional information has been added to Introduction in order to provide sufficient background to the study, as specified below. Accordingly, the relevant references have been added [15-31].

The information on environmental burden associated to Sb, in particular that concerning:

-  the release to water and accumulation in organisms

- the contextualization of the Sb pollution in Poland.

- the Sb oxidation states and  a related scheme (presently, the  Fig. 1. 

à The release of Sb to water and accumulation in organisms had been briefly described, based on the literature (P.3, L.64-77). 

à  Information on Sb pollution in Poland has also been introduced (P.2, L 51-53). Additionally, a comment to the value 427 mg/kg (the highest ever recorded from Poland) has been added, as suggested by Reviewer 1.

à Sb oxidation states have been presented in a related scheme (presently: the  Fig. 1, entitled:  Environmental transformations of antimony (Sb)  – a simplified scheme, after [17], modified.

à The hypothesis associated to the effects of the microbe-Sb interactions on Sb sorption or release from soils has been related to the literature (P.16, L. 400):

“… as confirmed recently by various authors [17, 18]”

3. As for the research design and methods – the Reviewer indicated that they “must be improved”. However, no specific comments nor suggestions were included in the Review. Therefore, no extensive changes have been made to this part of the manuscript except for those specified by Reviewers 1 and 2. Additionally, more detailed information on sampling sites has been introduced, to better characterize to edaphic  and environmental context of the study (P.5, L.149-152):

“The dumps ZS1, ZS2, DB, R and SG are surrounded by forests, while DM, sparsely covered by trees, is adjacent to fields and meadows. They are all situated in hilly or mountainous landscapes, typically in the valleys drained by the streams that continue through the villages and in some cases supply water to homesteads”.

Additionally, the description of methods has been made more clear by improvement of  English language and style.

4. Additional fragments of the texts have been introduced in the section “Results and discussion” in order to better explain the results in the context of environmental effects. In particular, a concluding statement related to environmental risk, has been added (P.16, L. 407-410):

“In certain conditions, in particular in the presence of biodegradable organic matter, antimony can be released into water in the amounts that might present a threat to the environment and to the health of local inhabitants.”

We hope that the results are now, after English proofreading, presented clearly enough.

Round  2

Reviewer 3 Report

Authors followed the recommendations; an additional minor revision is suggested in line 135, to replace “Those areas represented various geological settings, as described in another paper [50]” by “Geological settings and mining history of the five sampling sites were described by Karczewska et al. [50].

Author Response

Dear Reviewer,

Thank you for all your comments. Line 135 has been replaced by  “Geological settings and mining history of the five sampling sites were described by Karczewska et al. [50]."